# Hughes Abdominal Repair Trial (HART)—abdominal wall closure techniques to reduce the incidence of incisional hernias: feasibility trial for a multicentre, pragmatic, randomised controlled trial

Rhiannon L Harries,[1,2] Julie Cornish,[2,3] David Bosanquet,[1,2] Buddug Rees,[1] James Horwood,[1] Saiful Islam,[4] Nadim Bashir,[4] Alan Watkins,[4] Ian T Russell,[4] Jared Torkington,[1] on behalf of the HART Trial Management Group

[1]Department of Colorectal Surgery, University Hospital of Wales, Cardiff, UK
[2]Welsh Barbers Research Group, Cardiff, UK
[3]Department of Colorectal Surgery, Royal Glamorgan Hospital, Llantrisant, UK
[4]Swansea Trials Unit, Swansea University, Swansea, UK

**Correspondence to**
Professor Jared Torkington;
jared.torkington@wales.nhs.uk

## ABSTRACT

**Objectives** Incisional hernias are common complications of midline abdominal closure. The 'Hughes Repair' combines a standard mass closure with a series of horizontal and two vertical mattress sutures within a single suture. There is evidence to suggest this technique is as effective as mesh repair for the operative management of incisional hernias; however, no trials have compared Hughes repair with standard mass closure for the prevention of incisional hernia formation. This paper aims to test the feasibility of running a randomised controlled trial of a comparison of abdominal wall closure methods following midline incisional surgery for colorectal cancer, in preparation to a definitive randomised controlled trial.

**Design and setting** A feasibility trial (with 1:1 randomisation) conducted perioperatively during colorectal cancer surgery.

**Participants** Patients undergoing midline incisional surgery for resection of colorectal cancer.

**Interventions** Comparison of two suture techniques (Hughes repair or standard mass closure) for the closure of the midline abdominal wound following surgery for colorectal cancer.

**Primary and secondary outcomes** A 30-patient feasibility trial assessed recruitment, randomisation, deliverability and early safety of the surgical techniques used.

**Results** A total of 30 patients were randomised from 43 patients recruited and consented, over a 5-month period. 14 and 16 patients were randomised to arms A and B, respectively. There was one superficial surgical site infection (SSI) and two organ space SSIs reported in arm A, and two superficial SSIs and one complete wound dehiscence in arm B. There were no suspected unexpected serious adverse reactions reported in either arm. Independent data monitoring committee found no early safety concerns.

**Conclusions** The feasibility trial found no early safety concerns and demonstrated that the trial was acceptable to patients. Progression to the pilot and main phases of

### Strengths and limitations of this study

► This feasibility trial is not powered for a definitive study and simply reports the recruitment, deliverability and safety.

► We report blinded outcome data and have not included incisional hernia rates in order to prevent the introduction of bias or reduction of equipoise for future recruitment of the main trial.

► We acknowledge that randomising immediately prior to abdominal closure may increase the risk of selection bias into the trial; however, to overcome this we have collected information on the reasons why patients were not randomised after consenting in the screening log.

► The setting of the feasibility trial was chosen as the trial's lead site: a high-volume teaching hospital. This poses a potential limitation for the main trial at lower volume centres, in terms of ability to recruit participants over the time period (3 years).

► The Hughes Abdominal Repair Trial is a pragmatic trial, and as such we are allowing the control (mass closure) arm to be the responsible consultant surgeon's standard closure technique, and acknowledge that this may introduce a degree of variability in the control arm.

the trial has now commenced following approval by the independent data monitoring committee.

**Trial registration number** ISRCTN 25616490.

## INTRODUCTION

Incisional hernias are common complications of midline abdominal incisions, with a reported incidence of 12.8% at 2 years of follow-up in a systematic review of 14618 patients.[1] Within patients who have

undergone colorectal cancer resectional surgery, the rate of incisional hernia has been reported as high as 39.9%, including both open and laparoscopic approaches (40.9% and 37.1%, respectively).[2] They can result in significant morbidity and impaired quality of life,[3] and frequently require emergency surgery. Despite recent development in mesh technology, incisional hernia repair still has disappointingly high recurrence rates (up to 54% in suture repair and up to 36% in mesh repair).[4 5] Prevention of the development of incisional hernia therefore brings significant benefits for both patients and healthcare provision funding.

'Mass closure' remains the standard technique for abdominal closure (closing all layers of the abdominal wall, excluding the skin), with either non-absorbable or slow-resorbing sutures, such as polydioxanone (PDS).[6] A systematic review and meta-regression of over 14 000 patients found no difference in incisional hernia rate comparing suture material.[1] This poses the question as to whether improved suture technique may reduce incisional hernia formation. The STITCH trial,[7] a Dutch multicentre, randomised controlled trial, compared small-stitch continuous sutures with large-stitch standard mass closure in 560 patients. Results demonstrated a reduction in the rate of incisional hernia from 21% in the large bite group to 13% in the small bite group at 1-year follow-up. The CONTINT trial, currently still in recruitment, is comparing continuous with interrupted sutures in closing midline incisions after emergency laparotomy.[8]

The European Hernia Society Guidance on the closure of abdominal wall incisions (2015) recommended the use of prophylactic mesh augmentation for an elective midline laparotomy in a high-risk patient in order to reduce the risk of incisional hernia.[9] However, first, they determined that the evidence base for this was weak, and second in the UK mesh augmentation closure is infrequently used. It is for these reasons that it is still critical for other closure methods to be rigorously assessed for their role in incisional hernia prevention.

The eponymously titled 'Hughes Repair' (Professor Les Hughes, 1932–2011),[10] also known as the 'far-and-near' or 'Cardiff Repair',[11] combines a standard mass closure (two-loop 1 PDS sutures) with a series of horizontal and two vertical mattress sutures within a single suture (1 nylon), theoretically distributing the load along the incision length as well as across it (figure 1). The following are the principles:

1. To ensure, by palpation, that only sound normal tissues are used for the repair.
2. To use graduated tension for easy approximation.
3. Use a monofilament nylon suture, which has the advantage of slipping easily through tissues to create a pulley system.[12]

The Hughes repair has been shown to have outcomes as effective as the standard mesh repair in incisional hernia repairs.[13] It is also used for closing abdomens when patients are at high risk of incisional hernias,

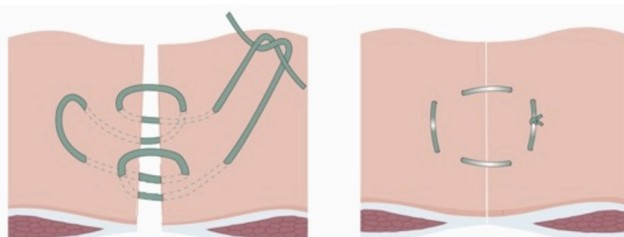

**Figure 1** Diagram showing the Hughes closure method, using a combination of standard mass closure with a series of horizontal and two vertical mattress sutures within a single suture. When the sutures are pulled to close the defect, the sutures lie both across and along the incision.

after complete abdominal wound dehiscence and laparostomy.[14]

This feasibility trial aimed to establish whether a randomised controlled trial to compare Hughes repair with standard mass closure for prevention of midline incisional hernia, in patients undergoing colorectal cancer resectional surgery, would be deemed acceptable to patients, achieve adequate recruitment and result in no early safety concerns.

## METHODS/DESIGN
### Study design
The Hughes Abdominal Repair Trial (HART) hypothesis is that the Hughes repair will reduce the incidence of clinically detected incisional hernia at 1 year in patients undergoing midline abdominal wall closure incisions following elective or emergency colorectal cancer surgery when compared with standard mass closure (figure 2). This is a 1:1 randomised controlled trial comparing two suture techniques for the closure of the midline abdominal wound following surgery for colorectal cancer.

### Setting and location
The feasibility trial took place at the trial's lead site University Hospital of Wales, Cardiff, a high-volume teaching hospital (1 of the 20 proposed recruitment sites for the main trial).

### Aims and outcome measures
The feasibility trial aimed to assess the ability of the trial to recruit and consent patients over a 5-month period and the deliverability and safety of the Hughes repair. The acceptability was assessed in terms of percentage of consenting versus refusing participants. Adequacy of recruitment is assessed in terms of number of recruited participants. Operation-specific adverse events (AEs) collected included surgical site infection and full wound dehiscence. Early surgical safety was assessed in terms of serious event and wound complication rates. In this paper we report blinded outcome data and have not included incisional hernia rates in order to prevent the introduction of bias or reduction of equipoise for future recruitment of the main trial.

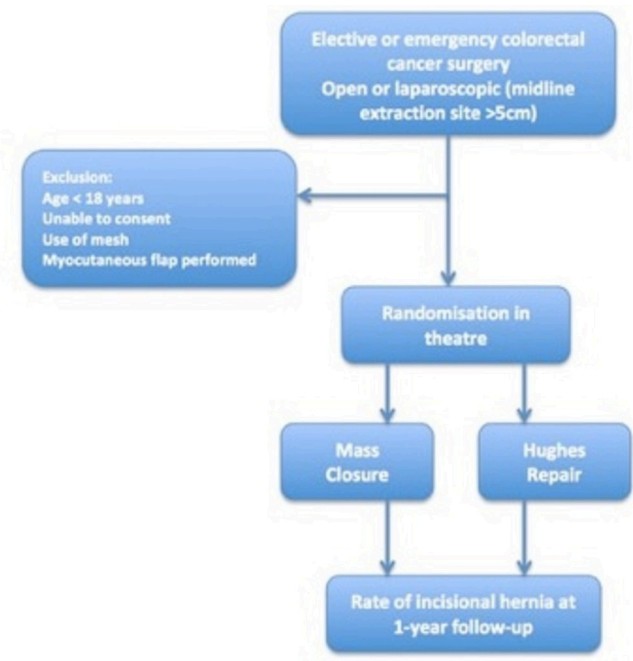

**Figure 2** Hughes Abdominal Repair Trial study design.

Independent data monitoring committee reviewed the unblinded safety data after completion of the feasibility phase. The outcome measures for the full trial have been previously reported,[15] with the primary outcome measure being the rate of incisional hernia at 1-year follow-up assessed by clinical examination. The full trial protocol can be accessed via the following link: https://njl-admin.nihr.ac.uk/document/download/2007245. Follow-up will continue for 5 years postoperatively; however, in this paper, only 12-month lost to follow-up data will be presented as the aim of this feasibility trial is to assess the deliverability and safety of the trial.

### Eligibility criteria

Eligibility criteria were assessed at two time points: at initial screening and at point of surgical closure/randomisation. Adult patients (aged 18 year or over), able to give informed consent, and undergoing either elective colorectal cancer surgery following full staging investigations including an abdominal CT scan or emergency surgery in those with a strong suspicion of colorectal cancer on abdominal CT scan were eligible at point of initial screening. All patients had to be suitable for either Hughes repair or standard mass closure. At point of surgical closure, eligibility was further assessed, and all patients who had a midline incision (open or laparoscopic assisted/converted) of 5 cm or more in length were deemed suitable for randomisation. Patients requiring mesh insertion or having an abdominal musculofascial flap for closure of the perineal defect in abdominoperineal wound closure were excluded.

### Consent

Patients were identified, approached and provided with a patient information leaflet. Consent for trial participation was gained by either consultant surgeons or surgical registrars who had current 'Good Clinical Practice' certification.

### Randomisation and data collection

An adaptive randomisation design was used to allocate eligible patients to groups of similar size. This randomisation is based on an independent, computer-based sequence, generated from an implementation of the dynamic algorithm, using operation category (elective or emergency) and surgeon as stratifying variables.[16] Patients were randomised in a 1:1 ratio to either mass closure or Hughes repair. Randomisation took place during surgery and as close as possible to the time when the surgeon commenced closure. During the feasibility trial, a telephone randomisation system was used. The patient was blinded to the treatment allocation assigned to them. Data management was supported by the Swansea Trials Unit.

### Surgical quality assurance

To assure the quality of the repair techniques, all surgeons participating in the trial (consultants and registrars) completed training and quality assessment on the Hughes repair. All participating surgeons were assessed by the chief investigator and were approved only when closure technique was satisfactory. A reference instructional video was provided to participating surgeons. To monitor the training of professionals contributing to HART, a log was maintained with the details of training, both surgical and in research governance notably 'Good Clinical Practice'. For the purposes of this pragmatic study, mass closure was taken to be the responsible consultant surgeon's standard closure technique.

### Radiological evaluation of incisional hernia

A dedicated trial radiologist determined whether there was a hernia present on the 1-year colorectal cancer surveillance CT scan. They defined an incisional hernia as herniation of the bowel or other intra-abdominal content outside the abdominal wall, and also identified the presence of other hernias and the quality of the recti muscle. All scans were performed using the standard departmental protocol for follow-up scans.

### Sample size

The feasibility trial aimed to recruit a total of 30 patients over a 5-month period, because the HART trial management group felt that such a sample size was indicative of the ability to recruit the sample proposed for the main trial within the established time frame. The sample size for the main study has been published previously.[14]

### Adverse events

An AE was defined as any untoward medical occurrence in a clinical trial participant to whom a study intervention has been administered and which does not necessarily have a causal relationship with this treatment. An AE can therefore be any unfavourable and unintended sign (including abnormal laboratory finding),

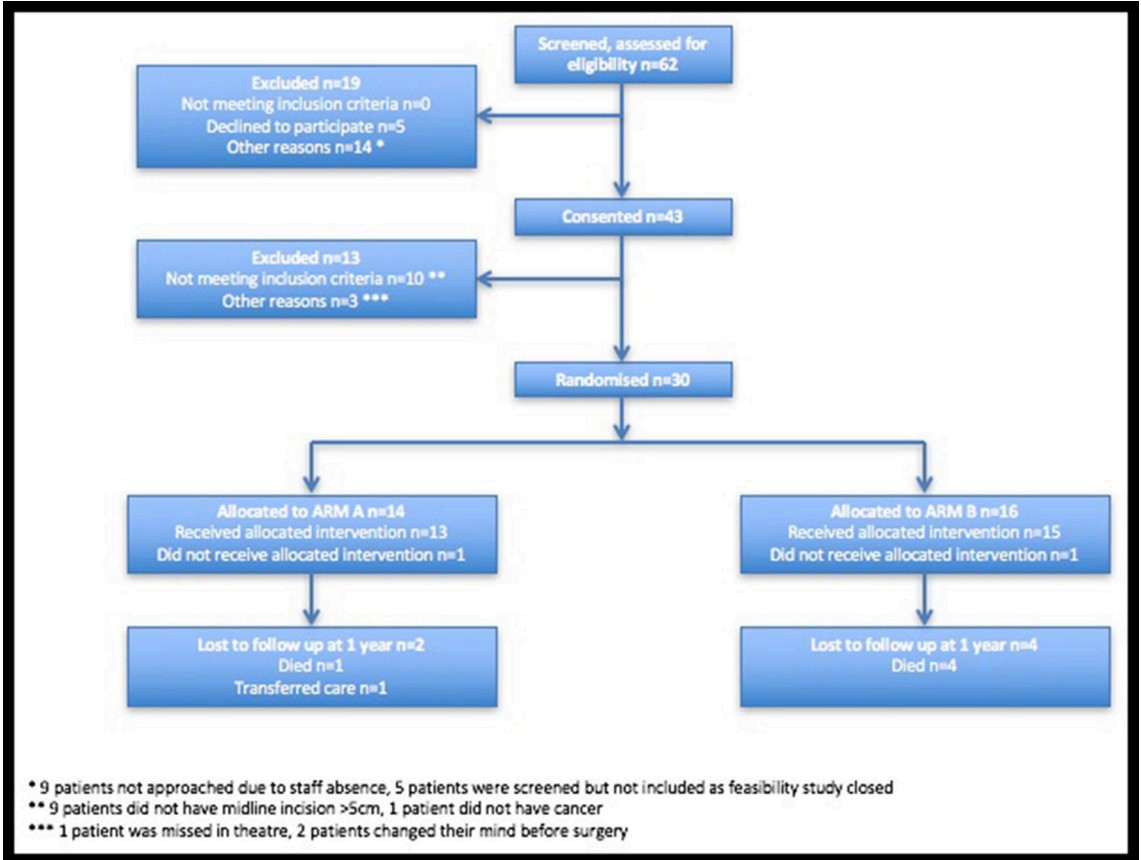

**Figure 3** Consolidated Standards of Reporting Trials diagram.

symptom or disease. The following are the AEs that are considered expected for patients undergoing colorectal surgery: lower respiratory tract infection, urinary tract infection, anastomotic leak, intra-abdominal sepsis, deep vein thrombosis, pulmonary embolus, wound infection, surgical site infection, wound breakdown, paralytic ileus, bleeding, myocardial infarction and stoma complications (prolapsed, retraction, dehiscence or hernia). However, if these events lead to death, that would be considered unexpected. These events may be classified as serious and will be recorded as such but will not require reporting to Research Ethics Committee. Additional information may be requested for AEs of special interest such as wound breakdown and surgical site infections.

A serious adverse event (SAE) is an adverse event that results in any of the following: death, was life-threatening, required hospitalisation or prolongation of existing hospitalisation, persistent or significant disability or incapacity, consists of a congenital anomaly or birth defect, or is otherwise considered medically significant by the investigator.

### Statistical analysis

A two-tailed Fisher's exact test was used to compare SAE rate between both arms. Differences were considered to be statistically significant at P≤0.05.

## RESULTS

### Recruitment and randomisation

A total of 62 patients were screened and assessed for eligibility for entry into the trial over a 5-month time period, October 2013 to February 2014 (figure 3). Of those screened, 43 patients consented to entry into the trial (69%). A total of 30 of the 43 patients were randomised in the operating theatre (14 patients were randomised to arm A and 16 patients were randomised to arm B). The reasons for exclusion are described in the Consolidated Standards of Reporting Trials diagram (figure 3).

For the 30 patients who were randomised, the median age was 74 years (IQR 66–78). There were 23 men and 7 women. Demographical data are presented in table 1.

In arm A, one patient had died prior to 12-month follow-up and a further patient had transferred care to another unit. In arm B, one patient had died prior to 12-month follow-up.

### Safety data

There were a total of 16 SAEs reported in 10 patients (table 2); SAE rate was 34% in arm A and 31% in arm B (P=1.0000). There were no suspected unexpected serious adverse reactions reported in either arm. With regard to wound-related complications, there was one superficial surgical site infection and two organ space surgical site infections reported in arm A, and two superficial surgical

**Table 1** Patient demographics and clinical characteristics

| | Arm A (n=14) | Arm B (n=16) | Total (n=30) |
|---|---|---|---|
| Gender, n (%) | | | |
| Male | 10 (71) | 13 (81) | 23 (77) |
| Female | 4 (29) | 3 (19) | 7 (23) |
| Median age (IQR) | 75 (61–78) | 73 (68–77) | 74 (66–78) |
| Mean BMI (Min–max) | 30 (22–49) | 29 (18–42) | 29 (18–49) |
| Smoker, n (%) | 1 (7) | 3 (19) | 4 (13) |
| Steroid/immuno suppression use, n (%) | 0 (0) | 0 (0) | 0 (0) |
| Diabetes, n (%) | 5 (36) | 4 (25) | 9 (30) |
| Connective tissue disorder, n (%) | 1 (7) | 0 (0) | 1 (3) |
| COPD, n (%) | 1 (7) | 2 (13) | 3 (10) |
| AAA (known or previous repair), n (%) | 0 (0) | 0 (0) | 0 (0) |
| Previous abdominal surgery, n (%) | 8 (57) | 8 (50) | 15 (50) |
| Neoadjuvant chemotherapy, n (%) | 1 (7) | 0 (0) | 1 (3) |
| Neoadjuvant radiotherapy, n (%) | 1 (7) | 1 (6) | 2 (7) |
| Incisional hernia present preoperatively, n (%) | 0 (0) | 1 (6) | 1 (3) |
| Previous incisional hernia repair, n (%) | 1 (7) | 0 (0) | 1 (3) |
| Non-incisional hernia present preoperatively, n (%) | 0 (0) | 3 (19) | 3 (10) |
| Mode of surgery, n (%) | | | |
| Laparoscopic | 4 (27) | 6 (38) | 10 (33) |
| Laparoscopic converted | 7 (50) | 3 (19) | 10 (33) |
| Open | 3 (21) | 7 (44) | 10 (33) |

AAA, abdominal aortic aneurysm; BMI, body mass index; COPD, chronic obstructive pulmonary disease.

site infections and one complete wound dehiscence requiring a return to theatre in arm B (table 3).

## DISCUSSION

The results of this feasibility trial demonstrated that a randomised controlled trial designed to compare two suture techniques for the prevention of midline incisional hernia, in patients undergoing cancer resectional surgery, was able to recruit 30 patients over 5 months, as planned. This suggests that the proposed sample size of 800 patients for the main full trial is achievable in the time scale with the proposed number of sites recruiting (approximately 20).[14]

The feasibility trial results established that the trial was acceptable to patients. Patient participation rates

**Table 2** Reported serious adverse events (SAEs)

| | Arm A | Arm B |
|---|---|---|
| Myocardial infarction | 2 | 2 |
| Lower respiratory tract infection | 2 | 1 |
| Pulmonary embolism | 1 | 0 |
| Renal failure | 0 | 1 |
| Anastomotic leak | 2 | 0 |
| Parastomal hernia | 0 | 1 |
| Superficial surgical site infection | 2 | 0 |
| Dehiscence | 0 | 1 |
| Death* | 1 | 0 |
| Total SAEs | **10** | **6** |
| Total patients affected | **5** | **5** |

*1 SAE reported was reported as 'death'; therefore, it had to be listed as an event of death. There were two other SAEs that resulted in death within the feasibility study.

were high, demonstrated by 69% of all eligible patients consenting to participation in the trial. Nine patients were screened for eligibility but not consented due to staff shortages, highlighting the importance of having adequate number of approved consenting staff on the delegation log. Due to the nature of the study, it was accepted that not all patients consented would eventually be randomised. In fact, there was a higher than expected number of patients consented and not eventually randomised (31%). Patients were consented but not randomised if the intraoperative procedure performed did not meet the inclusion criteria, for example not midline incision, conversion to open procedure using a non-midline incision or emergency patients found not to have a tumour intraoperatively. We acknowledge that this method may increase the risk of selection bias into the trial; however, to overcome this, we have collected information on the reasons why patients were not randomised after consenting (figure 3).

The setting of the feasibility trial was chosen as the trial's lead site, a high-volume teaching hospital. This poses a potential limitation for the main trial at lower volume centres, in terms of ability to recruit participants over the time period (3 years). However, the sample size required for the pilot and main trial is 800 across roughly

**Table 3** Wound-related complications

| | Arm A | Arm B |
|---|---|---|
| Superficial SSI | 1 | 2 |
| Deep SSI | 0 | 0 |
| Organ space SSI | 2 | 0 |
| Wound dehiscence | 0 | 1 |
| Total wound-related complications | 3 | 3 |

The data monitoring committee reviewed the unblinded adverse events data and identified no safety concerns.
SSI, surgical site infection.

20 sites, which equates to 40 participants required per site and should be achievable over the trial time period, even for lower volume centres, particularly given the incidence of colorectal cancer within the UK.

The HART trial is a pragmatic trial, and as such we are allowing the control (mass closure) arm to be the responsible consultant surgeon's standard closure technique. We acknowledge that this may introduce a degree of variability in the control arm, but in order to counter this our sample size for the main trial has been powered to be able to stratify for site and surgeon.

The SAE rate and wound-related complications were similar between both arms, and reassuringly there were no suspected unexpected serious adverse reactions reported. It is anticipated that reporting on the full trial will take place in 2019.

**Contributors** RLH, JC, DB, BR, ITR, JT conceived, designed and drafted the initial protocol. JH aided in redrafting and revising the protocol and contributed heavily to completion of the feasibility trial and pilot stages. SI, AW, NB, ITR have written and designed the analysis plan and revised the protocol. JC, DB, ITR, JT led the team that acquired the funding. RLH wrote the first draft of the manuscript. All authors agree to be accountable for all aspects of the work. All authors read and approved the final manuscript.

**Funding** The NIHR Health Technology Assessment programme requested this feasibility trial as a prerequisite for awarding grant 12/35/29 for the pilot and main phases of HART.

**Competing interests** None declared.

**Ethics approval** The study was approved by the Wales Research Ethics Committee (MREC 12/WA/0374).

**Provenance and peer review** Not commissioned; externally peer reviewed.

**Data sharing statement** No additional data available.

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
