## [Reviewer comments · BMJ Open]

ARTICLE DETAILS

TITLE (PROVISIONAL)	Hughes Abdominal Repair Trial (HART) - Abdominal wall closure techniques to reduce the incidence of incisional hernias: Feasibility trial for a multi-centre pragmatic randomised controlled trial
AUTHORS	Harries, Rhiannon; Cornish, Julie; Bosanquet, David; Rees, Buddug; Horwood, James; Islam, Saiful; Bashir, Nadim; Watkins, Alan; Russell, Ian; Torkington, Jared

VERSION 1 – REVIEW

REVIEWER	GH van Ramshorst NKI/Dutch Cancer Institute and VU Medical Center
REVIEW RETURNED	14-Apr-2017

GENERAL COMMENTS	Thank you for the opportunity to review this interesting paper. Generally, the paper is well written and easy to read. The authors should be congratulated on completing the feasibility study so quickly and the overall interest in the topic. With regard to the description of the background of the clinical topic, methodology and discussion, I have some concerns. Title page There seems to be a high number of authors for this paper. Do all authors qualify for authorship and/or could be included as HART Trial Management Group reference? Acquiring funding alone does not qualify for authorship, but does for collaboratorship I believe. Introduction Line 6, 7: The incidence of incisional hernias is quite higher in other publications. I would suggest to refer to other papers as well, possibly targeted at your study population in particular Line 19-37: It is suggested to refer to the Milbourn et al and other Israelsson group data too and to the EHS guidelines for closure of the abdominal wall. Line 40-cont: It would be good to explain the differences between the Hughes repair and the continuous double loop closure technique as described by Luijendijk et al. Methods/Design Line 49-cont: It is strongly recommended to extend the inclusion criteria to patients undergoing midline laparotomy for other indications than colorectal cancer alone. Also, it is recommended to only use this technique in patients who are deemed unfit for laparoscopic surgery, as this is the current standard in colorectal surgery. It is recommended to exclude patients who undergo laparoscopically assisted procedures, as well as patients with incisional hernia at the time of surgery.
--

The length criterium of 5 cm seems to suggest that patients with extraction sites were deemed suitable for inclusion? There is no reference to previous (midline) abdominal surgery.
There is no description of standard mass closure. The standard mass closure with the best results thus far is that as described by Deerenberg et al and Millbourn et al.
The size of the cohort seems relatively small to find adverse events. Could the authors describe in some more detail how was the group size determined, instead of only referring to reference nr. 13? Was it considered to randomize patients e.g. 2:1?
Surgical Quality Assurance: The sentence "For the purposes of the study, mass closure was taken to be the responsible consultant surgeon's standard closure technique", appears to ignore some essential aspects of surgical closure: type of suture/needle, suture length to wound length ratio, bite size, intersuture distance, etc. It is recommended to dedicate a paragraph in the methods to the description of both techniques.
Randomisation: please explain in more detail the process of randomisation (envelope, web-based, etc)
Statistics: It would be appropriate to compare both groups and the results if you report a control arm.

Results

With this number of patients, percentages should be rounded off to the nearest whole number (e.g., 31% instead of 31,25%).
Line 36: infections instead of infection (n=2)
Were both groups comparable?

Discussion

The discussion appears to be relatively short. There are a few concerns which deserve to be discussed:

1. There were few patients included in both arms
2. Is the standard technique performed well, especially with one full wound dehiscence in arm B? If not, why not improve on this technique first before entering a randomized controlled study? I am very concerned that the final publication of your main study will prove very difficult if your control arm is not the 'best' proven technique.
3. How will the quality of the closure in both arms be assured? How do we know the amount of tension on the suture, or the number of sutures placed in both techniques (i.e. suture length to wound length ratio?)
4. There is no mentioning of high risk groups (e.g., obesity, aortic aneurysms, previous hernias, etc)
5. There is no mentioning of the decreasing (relative) proportion of patients undergoing open elective colorectal surgery
6. There is no mentioning of the decreasing numbers of patients undergoing emergency surgery
7. There is no mentioning of alternatives for prevention of incisional hernia (e.g., prophylactic meshes)

Table 1:

- round off percentages
- how was connective tissue disorder defined?
- preoperative incisional hernia (repair) should disqualify a patient for participation in the future trial.
- laparoscopic surgery should disqualify for participation
- were there significant differences between both groups?

	Table 3: -does not add more information (as partially copied from table 2) Figure 3: -It is recommended to change 'staff holiday' into 'staf absence' In summary, it would be very interesting to see the final results of this study in a few years' time. The exact control arm and experimental arm should be clearly defined and the quality of both techniques should be assured. For now, I am afraid that the paper needs some major changes and cannot be published in the current form.
--	---

REVIEWER	rajeev Sinha Department of Surgery MLB Medical College Jhansi UP INDIA
REVIEW RETURNED	21-May-2017

GENERAL COMMENTS	A feasibility study should be followed by a randomized prospective study comparing this method with the outcomes of laparoscopic incisional hernia repair as well as with the others trials mentioned especially the dutch 'STITCH' trial of the open method of repair
---

REVIEWER	Fabio Cesare Campanile Ospedale "San Giovanni Decollato - Andosilla" AUSL VT Civita Castellana, VT Italy
REVIEW RETURNED	17-Jun-2017

GENERAL COMMENTS	1. Title and abstract: The authors have met the CONSORT "item 1a" requirements (Title and abstract). However, they may want to use the word "trial" instead of study, as recommended by the guideline (as in "multi-centre randomised feasibility trial"), but this is certainly not necessary. Even if it can be inferred from the abstract conclusions, it could be appropriate to include, within the context of the abstract, a sentence explaining that the study is in preparation to a definitive RCT. The abstract should also include a description of the areas of uncertainty to be addressed and a statement of the feasibility aims and objectives of the study, instead of the generic "This paper aims to test the feasibility of running a randomised controlled trial..." 2. Methods (objectives and measurements): The objectives of the study are correctly listed at the end of the methods section (item 2b) as: "...acceptable to patients, achieve adequate recruitment and result in no early safety concerns". However, in the "methods" section, when it comes to define the prespecified "assessments or measurements to address each feasibility trial objective specified in item 2b" (item 6a), the measurement for "acceptability" and "adequacy" of recruitment are not fully defined (for example, it could be: "acceptability is assessed in terms of percentage of consenting vs refusing participants" or "adequacy of recruitment is assessed in terms of number of recruited participants").
--

The measures should address each separate feasibility research question, otherwise it may be impossible to evaluate the statistical characteristics of the study and only a qualitative outcome description is achievable. Also, in the “aims and outcome measures” paragraph, the objective “deliverability” of the Hughes Repair is introduced, but it was not mentioned in the introduction among the aims of the trial.

Measurements for surgical safety are, also, described in generic terms (“post-operative complications, serious adverse event reporting and wound diaries”) and not as an appropriate variable (for example: morbidity, total postoperative complication rate, operation specific morbidity rate, infection rate...).

3. Methods (sample size): Item 7a of the cited CONSORT guidelines recommend that the rationale for the numbers chosen for the feasibility trial be specified in the methods section; the manuscript “sample size” section specifies only the recruitment aim and not its rationale. In a feasibility trial, of course, the rationale may be somewhat generic or, at least, not as well defined as required for the final RCT, but still it should be mentioned. In this particular case, it could be related to the ability to recruit the sample proposed for the main trial (800, as briefly mentioned only in the discussion) in a certain amount of time.

4. Methods (statistical methods): Item 9 recommend that not only the type of randomisation, but also the mechanism to produce the random sequence should be specified. Similarly, item 10 asks who generated it.

The explanation of the statistical methods used to compare groups (item 12a) is entirely missing from the methods section.

5. Methods (setting and location): Also, setting and location (item 4b) are not reported; this aspect is particularly important in view of the multicentre organization of the main trial (has the feasibility study been carried out in one of the 20 proposed recruitment sites? Has this site characteristics that are representative of the rest of the sites? For example, a high volume centre could be facilitated in recruiting the feasibility sample, but lower volume sites could slow down the recruitment during the main trial; acceptability could be different between academic and non-academic centres). Some of these features could be also part of the study limitation to be detailed in the discussion.

6. Results: Results should be presented according to the assessments or measurements specified, in the methods section, for each feasibility trial objective (item 17a)

Also, dates of recruitment and follow-up (item 14a) are missing.

Table 2 lists the “serious” adverse events, but a definition of “serious” is not given. The same applies for the definition of “unexpected” Vs. “expected” event.

7. The particular design of the trial includes both laparoscopic (converted or not) and open surgery; the eligibility criteria includes the presence of a minimum 5 cm incision. Then, in the same group we have patients with 5 (non-converted laparoscopy) and, probably, 25 cm (open surgery) incisions. Table 1 includes data about the mode of surgery (laparoscopic or not). No data about the length of incision is included in the table or text. Therefore, we cannot evaluate the presence of significant differences between the study and control group on this relevant aspect.

	Of course, the evaluation of the intervention effects (ventral hernia rate) is not important in the feasibility trial and, correctly, is not among its objectives. However, this problem will have to be faced in the main trial. 8. Discussion: there are no “limitations” and “generalisability” sections. Also, it would be interesting to know if there are any “implications for progression from pilot to future definitive trial, including any proposed amendments” (item 22a). For example: do the author think that the choice to randomise patients as close as possible to the abdominal wall closure (instead of at the beginning of the operation) is actually the best choice? Could it introduce any selection bias? 9. CONSORT checklist: the reported CONSORT checklist includes informations that I was unable to find in the manuscript. For example at item 24 is declared that page 6 specifies about where the full trial protocol can be accessed, but on page 6 I could not find that information. Items 20 and 21 indicate page 10 for the limitation and generalisability sections but, as previously noted, those sections are not present. Page 7 should include informations about the method used to generate the random allocation sequence, the mechanism used to implement the random allocation sequence, who generated the random allocation sequence, who enrolled participants, and who assigned participants to interventions (items 8-10), but those details are actually not available. In many other instances, the information provided in the checklist are not fully consistent with the manuscript. In conclusion I believe that the paper is interesting but some revisions are essential to better explain the statistical methods and adhere to the reporting standards.
--	---

REVIEWER	Dr Victoria Allgar University of York
REVIEW RETURNED	12-Jul-2017
GENERAL COMMENTS	A simple paper reporting the feasibility outcomes of this study. Descriptive statistics are presented and are appropriate.

VERSION 1 – AUTHOR RESPONSE

Reviewer: 1

Reviewer Name: GH van Ramshorst

Institution and Country: NKI/Dutch Cancer Institute and VU Medical Center

Please state any competing interests: None declared

Please leave your comments for the authors below

General comment: Thank you for the opportunity to review this interesting paper. Generally, the paper is well written and easy to read. The authors should be congratulated on completing the feasibility study so quickly and the overall interest in the topic. With regard to the description of the background of the clinical topic, methodology and discussion, I have some concerns.

Title page

Comment: There seems to be a high number of authors for this paper. Do all authors qualify for authorship and/or could be included as HART Trial Management Group reference? Acquiring funding alone does not qualify for authorship, but does for collaboratorship I believe.

Response: Our authorship has been revised accordingly

Introduction

Comment: Line 6, 7. The incidence of incisional hernias is quite higher in other publications. I would suggest to refer to other papers as well, possibly targeted at your study population in particular. The referenced paper is a systematic review and meta-regression which contains 14,618 patients. The paper contains more patients than other large meta-analysis on the subject (Diener et al 7711 patients and van't Riet et al 6566 patients). We have changed the wording to 'Incisional hernias are common complications of midline abdominal incisions, with a reported incidence of 12.8% at 2 years follow-up in a systematic review of 14,618 patients.' And have added 'Within patients who have undergone colorectal cancer resectional surgery, the rate of incisional hernia has been reported as high as 39.9%, including both open and laparoscopic approaches, (40.9% and 37.1%, respectively)' Line 19-37: It is suggested to refer to the Milbourn et al and other Israelsson group data too and to the EHS guidelines for closure of the abdominal wall.

Response: Milbourn data has already been discussed in the introduction (STITCH trial) A paragraph has been added to the introduction from the EHS guidelines.

Comment: Line 40-cont. It would be good to explain the differences between the Hughes repair and the continuous double loop closure technique as described by Luijendijk et al.

Response: There is a figure describing Hughes Repair within the paper. The control arm represents modern everyday surgical practice in each setting and hence was not subject to the quality assurance of the intervention arm and we therefore feel a description is unnecessary.

Methods/Design

Comment: Line 49-cont. It is strongly recommended to extend the inclusion criteria to patients undergoing midline laparotomy for other indications than colorectal cancer alone.

We have included only patients undergoing midline incisional surgery for colorectal cancer resection as one of the tertiary outcomes in the main trial is CT detection of Incisional hernias as compared to clinical examination. In the UK, colorectal cancer patients undergo CT scanning as routine for their cancer surveillance therefore not subjecting them to further radiation.

Response: On a separate note our main trial commenced in 2014 and is due to complete within the next 4 months. Therefore, there will be no changes made to the main trial at this stage.

Comment: Also, it is recommended to only use this technique in patients who are deemed unfit for laparoscopic surgery, as this is the current standard in colorectal surgery. It is recommended to exclude patients who undergo laparoscopically assisted procedures, as well as patients with incisional hernia at the time of surgery.

Response: Patients undergoing midline specimen extraction as part of laparoscopic colorectal surgery still can result in incisional hernia formation. Pereira et al found an incisional hernia rate of 37% following laparoscopic colorectal resectional surgery. This is why we feel it necessary to include these patients. This is a pragmatic study and therefore we feel it important to include patients who present with incisional hernia at the time of surgery. As reiterated above no changes will be made to the main trial at this stage. We will however take on board the comments when reporting on our main trial and consider sub group analysis based on this.

Comment: The length criterium of 5 cm seems to suggest that patients with extraction sites were deemed suitable for inclusion? There is no reference to previous (midline) abdominal surgery.

Response: Previous midline abdominal surgery was not an exclusion criteria. Exclusion criteria is listed on page 6/7.

Comment: There is no description of standard mass closure. The standard mass closure with the best results thus far is that as described by Deerenberg et al and Millbourn et al.

Response: Our trial is a pragmatically designed trial and therefore our protocol states 'For the purposes of the study mass closure will be taken to be the responsible consultant surgeon's standard closure technique.' This is stated in the methods section> surgical quality assurance section.

Comment: The size of the cohort seems relatively small to find adverse events. Could the authors describe in some more detail how was the group size determined, instead of only referring to reference nr. 13? Was it considered to randomize patients e.g. 2:1?

Response: Randomisation 1:1 ratio has been chosen for this trial and 2:1 ratio was not considered.

Comment: Surgical Quality Assurance: The sentence "For the purposes of the study, mass closure was taken to be the responsible consultant surgeon's standard closure technique", appears to ignore some essential aspects of surgical closure: type of suture/needle, suture length to wound length ratio, bite size, intersuture distance, etc. It is recommended to dedicate a paragraph in the methods to the description of both techniques.

Response: It is a pragmatic study so therefore the choose of suture/needle, bite size etc were at the choose of the consultant surgeon. The wording has been changed to reflect this 'For the purposes of this pragmatic study, mass closure was taken to be the responsible consultant surgeon's standard closure technique'

Comment: Randomisation: please explain in more detail the process of randomisation (envelope, web-based, etc)

Response: We have added in the line 'During the feasibility trial, a telephone randomisation system was used.'

Comment: Statistics: It would be appropriate to compare both groups and the results if you report a control arm.

Response: The outcomes of the feasibility trial were to report on the acceptability to patients, assess if adequate recruitment was possible and to ensure no early safety concerns. Comparing demographic data between both groups is therefore unnecessary. We have however added in a fishers exact test to compare SAE rate between both arms.

Results

Comment: With this number of patients, percentages should be rounded off to the nearest whole number (e.g., 31% instead of 31,25%).

Response: This has been corrected

Comment: Line 36. infections instead of infection (n=2)

Response: This has been corrected

Discussion

The discussion appears to be relatively short. There are a few concerns which deserve to be discussed:

1. There were few patients included in both arms

Response: The HART trial management group felt that the sample size for the feasibility trial was adequate, and safety was assessed by the independent data monitoring committee who raised no concerns.

2. Is the standard technique performed well, especially with one full wound dehiscence in arm B? If not, why not improve on this technique first before entering a randomized controlled study? I am very concerned that the final publication of your main study will prove very difficult if your control arm is not the 'best' proven technique.

Response: We thank the reviewer for the comments. As stated previously we have almost completed recruitment to our main trial and therefore changes cannot be made at this stage.

3. How will the quality of the closure in both arms be assured? How do we know the amount of tension on the suture, or the number of sutures placed in both techniques (i.e. suture length to wound length ratio?)

Response: This is a very valid comment and applies to all trials of surgical technique. Surgeons required sign off, of their technique prior to being able to join the trial. The control arm represents modern everyday surgical practice in each setting and hence was not subject to the quality assurance of the intervention arm.

4. There is no mentioning of high risk groups (e.g., obesity, aortic aneurysms, previous hernias, etc)

Response: This data was collected routinely and will allow high risk group identification at analysis in the main trial.

5. There is no mentioning of the decreasing (relative) proportion of patients undergoing open elective colorectal surgery

Response: This is accepted and not a concern as we included laparoscopic patients that a) had a midline extraction site or b) were converted to open surgery.

6. There is no mentioning of the decreasing numbers of patients undergoing emergency surgery

Response: We have no evidence to support or refute this at present in the UK and is not of major relevance to this trial.

7. There is no mentioning of alternatives for prevention of incisional hernia (e.g., prophylactic meshes)

Response: The following has been added to the introduction 'European Hernia Society Guidance on the closure of abdominal wall incisions (2015) recommended the use of prophylactic mesh augmentation for an elective midline laparotomy in a high-risk patient in order to reduce the risk of incisional hernia. However they determined that the evidence base for this was weak. Hence why we feel it is critical for other closure methods be rigorously assessed for their role in incisional hernia prevention.'

Table 1 Comments:

-round off percentages

This has been corrected

-how was connective tissue disorder defined?

This has not been defined in the trial protocol but assumes the medical diagnosis of an appropriately named connective tissue disorder which was recorded at baseline.

-preoperative incisional hernia (repair) should disqualify a patient for participation in the future trial.

This is not within our exclusion criteria. No changes will be made to the main trial at this stage as we are nearly close to completing recruitment. A previous incisional hernia may place the patient in a high risk group and we felt it vital that this subjects were included

-laparoscopic surgery should disqualify for participation

Please see above comments

Table 3.

Comment: does not add more information (as partially copied from table 2)

Response: We feel its important to highlight the wound related complications in a separate table. It also contains other information that isn't captured in Table 2.

Figure 3

Comment: It is recommended to change 'staff holiday' into 'staff absence'

Response: This has been corrected

In summary, it would be very interesting to see the final results of this study in a few years' time. The exact control arm and experimental arm should be clearly defined and the quality of both techniques should be assured. For now, I am afraid that the paper needs some major changes and cannot be published in the current form.

Reviewer: 2

Reviewer Name: Rajeev sinha

Institution and Country: Department of Surgery, MLB Medical College, Jhansi, UP, INDIA

Please state any competing interests: None declared

Please leave your comments for the authors below

Comment: A feasibility study should be followed by a randomized prospective study comparing this method with the outcomes of laparoscopic incisional hernia repair as well as with the others trials mentioned especially the dutch 'STITCH' trial of the open method of repair

Response: Thank you for the reviewers comments. At this stage our trial is looking at primary abdominal wall closure rather than incisional hernia repair. As did the STITCH trial. Of course this does raise an interesting consideration for future research

Reviewer: 3

Reviewer Name: Fabio Cesare Campanile

Institution and Country: Ospedale "San Giovanni Decollato - Andosilla", AUSL VT, Civita Castellana, VT, Italy

Please state any competing interests: none declared

Please leave your comments for the authors below

The authors present a manuscript about a well conducted randomised feasibility trial.

I have used the "CONSORT 2010 statement extension to randomised pilot and feasibility trials" to evaluate the manuscript.

Comment 1. Title and abstract: The authors have met the CONSORT "item 1a" requirements (Title and abstract). However, they may want to use the word "trial" instead of study, as recommended by the guideline (as in "multi-centre randomised feasibility trial"), but this is certainly not necessary.

This has been corrected throughout

Even if it can be inferred from the abstract conclusions, it could be appropriate to include, within the context of the abstract, a sentence explaining that the study is in preparation to a definitive RCT.

This has been added in to the abstract

The abstract should also include a description of the areas of uncertainty to be addressed and a statement of the feasibility aims and objectives of the study, instead of the generic "This paper aims to test the feasibility of running a randomised controlled trial...".

Response: We have elaborated in the main paper but the word count limits this in the abstract

Comment 2. Methods (objectives and measurements): The objectives of the study are correctly listed at the end of the methods section (item 2b) as: "...acceptable to patients, achieve adequate recruitment and result in no early safety concerns". However, in the "methods" section, when it comes to define the prespecified "assessments or measurements to address each feasibility trial objective specified in item 2b" (item 6a), the measurement for "acceptability" and "adequacy" of recruitment are not fully defined (for example, it could be: "acceptability is assessed in terms of percentage of consenting vs refusing participants" or "adequacy of recruitment is assessed in terms of number of recruited participants"). The measures should address each separate feasibility research question, otherwise it may be impossible to evaluate the statistical characteristics of the study and only a qualitative outcome description is achievable.

Response: This has been added to the methods section

Comment: Also, in the “aims and outcome measures” paragraph, the objective “deliverability” of the Hughes Repair is introduced, but it was not mentioned in the introduction among the aims of the trial. Deliverability includes the ability to recruit patients into the trial as mentioned in the introduction. Measurements for surgical safety are, also, described in generic terms (“post-operative complications, serious adverse event reporting and wound diaries”) and not as an appropriate variable (for example: morbidity, total postoperative complication rate, operation specific morbidity rate, infection rate...).

Response: We have reported in the paper the outcomes that were required by the Data Monitoring Committee as per the aims of the feasibility trial, and is similar to how other feasibility trials publications report their outcomes.

Comment 3. Methods (sample size): Item 7a of the cited CONSORT guidelines recommend that the rationale for the numbers chosen for the feasibility trial be specified in the methods section; the manuscript “sample size” section specifies only the recruitment aim and not its rationale. In a feasibility trial, of course, the rationale may be somewhat generic or, at least, not as well defined as required for the final RCT, but still it should be mentioned. In this particular case, it could be related to the ability to recruit the sample proposed for the main trial (800, as briefly mentioned only in the discussion) in a certain amount of time.

Response: The HART trial management group felt that the sample size for the feasibility trial was adequate, and safety was assessed by the independent data monitoring committee who raised no concerns.

Comment 4. Methods (statistical methods): Item 9 recommend that not only the type of randomisation, but also the mechanism to produce the random sequence should be specified. Similarly, item 10 asks who generated it.

We have added the following to the methods section ‘An adaptive randomisation design was used to allocate eligible patients to groups of similar size [reference 16].’ and ‘ During the feasibility trial, a telephone randomisation system was used.’

The explanation of the statistical methods used to compare groups (item 12a) is entirely missing from the methods section.

Response: This has been added to the methods section

Comment 5. Methods (setting and location): Also, setting and location (item 4b) are not reported; this aspect is particularly important in view of the multicentre organization of the main trial (has the feasibility study been carried out in one of the 20 proposed recruitment sites? Has this site characteristics that are representative of the rest of the sites? For example, a high volume centre could be facilitated in recruiting the feasibility sample, but lower volume sites could slow down the recruitment during the main trial; acceptability could be different between academic and non-academic centres). Some of these features could be also part of the study limitation to be detailed in the discussion.

Response: This has been added to the methods section

Comment 6. Results: Results should be presented according to the assessments or measurements specified, in the methods section, for each feasibility trial objective (item 17a). Also, dates of recruitment and follow-up (item 14a) are missing.

Response: This has been added to the results section

Comment: Table 2 lists the “serious” adverse events, but a definition of “serious” is not given.

Response: This has been added to methods section

Comment: The same applies for the definition of “unexpected” Vs. “expected” event.

Response: This has been added to methods section

Comment 7. The particular design of the trial includes both laparoscopic (converted or not) and open surgery; the eligibility criteria includes the presence of a minimum 5 cm incision. Then, in the same group we have patients with 5 (non-converted laparoscopy) and, probably, 25 cm (open surgery) incisions. Table 1 includes data about the mode of surgery (laparoscopic or not). No data about the length of incision is included in the table or text. Therefore, we cannot evaluate the presence of significant differences between the study and control group on this relevant aspect. Of course, the evaluation of the intervention effects (ventral hernia rate) is not important in the feasibility trial and, correctly, is not among its objectives. However, this problem will have to be faced in the main trial.

Response: We thank the reviewer for highlighting considerations for reporting of our main trial results. Data is being captured on the intraoperative case report form on length of midline skin incision. We can therefore present data / analysis as per wound length. As the reviewer correctly identifies the incisional hernia rate is not important in the feasibility trial and therefore not amongst the objectives of added to the results.

Comment 8. Discussion: there are no “limitations” and “generalisability” sections. Also, it would be interesting to know if there are any “implications for progression from pilot to future definitive trial, including any proposed amendments” (item 22a). For example: do the author think that the choice to randomise patients as close as possible to the abdominal wall closure (instead of at the beginning of the operation) is actually the best choice? Could it introduce any selection bias?

Response: We believe it is vital to randomise as close as possible to the abdominal wall closure due to the fact that a significant number of patients who have been consented will end up not having a midline incision of 5cms i.e. Those who have been laparoscopically converted using a transverse/pfannelsteil incision or intraoperative decision to use mesh augmentation for closure. We acknowledge that there maybe a risk of selection bias due to this method however to overcome this information has been collected on the reasons why patients were not randomised after consenting. As you can see in figure 3, it details that 9 patients were not randomised due to lack of midline incisional of 5cms or more and 1 patient due to not having cancer discovered intraoperatively.

Comment 9. CONSORT checklist: the reported CONSORT checklist includes informations that I was unable to find in the manuscript. For example at item 24 is declared that page 6 specifies about where the full trial protocol can be accessed, but on page 6 I could not find that information.

Response: This has been added

Comment: Items 20 and 21 indicate page 10 for the limitation and generalisability sections but, as previously noted, those sections are not present.

Response: This has been added

Comment: Page 7 should include informations about the method used to generate the random allocation sequence, the mechanism used to implement the random allocation sequence, who generated the random allocation sequence, who enrolled participants, and who assigned participants to interventions (items 8-10), but those details are actually not available.

Response: See above

Comment: In many other instances, the information provided in the checklist are not fully consistent with the manuscript.

In conclusion I believe that the paper is interesting but some revisions are essential to better explain the statistical methods ad adhere to the reporting standards.

Reviewer: 4

Reviewer Name: Dr Victoria Allgar

Institution and Country: University of York

Please state any competing interests: None declared

Please leave your comments for the authors below

Comment: A simple paper reporting the feasibility outcomes of this study. Descriptive statistics are presented and are appropriate.

Response: We thank the fourth reviewer for their comments.

VERSION 2 – REVIEW

REVIEWER	Fabio Cesare Campanile, MD, FACS Ospedale San Giovanni Decollato "Andosilla" . AUSL VT Civita Castellana Italy
REVIEW RETURNED	26-Aug-2017

GENERAL COMMENTS	The authors should be congratulated for their prompt review of the manuscript. They modified most of the areas of concern. The paper is now more complete and interesting. However, I realized that my comments have not always been clear. I am sorry I was not able to correctly explain the meaning of my observations. This is particularly true in the methods section. In general, when I expressed a concern in this section, it was not due to a doubt about the results of the study or that I had concerns about its safety, but I simply meant that that characteristic (required by the CONSORT guideline) was not clearly expressed in the methods section.
--

For example:

1. "Measurements for surgical safety are, also, described in generic terms ("post-operative complications, serious adverse event reporting and wound diaries") and not as an appropriate variable (for example: morbidity, total postoperative complication rate, operation specific morbidity rate, infection rate...):" The comment was not conveying the idea that measures of surgical safety were not adequate, and of course you correctly reported the outcomes required by the Monitoring Data Committee. I only pointed out that a correct report of them did not correspond a specific notation in the methods section. The authors could increase the value of their paper by specifying the variable adopted for the outcome measurement instead of (or in addition to) the font of data they used. In particular, in the result section, the outcome is expressed in terms of "serious event rate" and "wound complication rate"; because the authors have now well explained the definition of serious event, in the methods section they could write "Early surgical safety was assessed in terms of serious event and wound complication rates" or "Early surgical safety was assessed using post-operative complication reports, serious adverse event reporting and wound diaries, as wound complication and serious event rate".

2. "Item 7a of the cited CONSORT guidelines recommend that the rationale for the numbers chosen for the feasibility trial be specified in the methods section; the manuscript "sample size" section specifies only the recruitment aim and not its rationale. [...] In this particular case, it could be related to the ability to recruit the sample proposed for the main trial (800, as briefly mentioned only in the discussion) in a certain amount of time." Of course, as you specified in your reply, the HART trial management group felt that the sample size for the feasibility trial was adequate. But this aspect needs to be detailed in the methods section. For example: "The feasibility trial aimed to recruit a total of 30 patients over a 5-month period, because the HART trial management group felt that such a sample size was indicative of the ability to recruit the sample proposed for the main trial within the established time frame"

3. "the mechanism to produce the random sequence should be specified": besides the randomization design you should mention the mechanism to produce the random sequence (i.e. closed envelopes, randomization utility on a computer, coin toss). Also who generated the sequence (a surgeon, a statistician, a nurse?) and who assigned the patient to the study (it seems the main surgeon after having seen the operative picture) are useful details here. These apparently trivial details greatly enhance the value of the report.

4. In the results section: "Also, dates of recruitment and follow-up (item 14a) are missing": "This has been added to the results section". The data of recruitment has been added. Still, I cannot find if there was a minimum follow up period granted to all patients and the average follow up for each group. A mention of patients eventually lost to follow up would also be nice. Of course, a trial significance is as good as its follow up rates, and adding these data would also have a positive impact on the readers.

In conclusion I think that the manuscript can be accepted for publication on BMJ Open, and I also think that adding those minor details to the final text would enhance the value of the paper.

VERSION 2 – AUTHOR RESPONSE

Reviewer: 3

Reviewer Name: Fabio Cesare Campanile, MD, FACS

Institution and Country: Ospedale San Giovanni Decollato "Andosilla" . AUSL VT, Civita Castellana, Italy

Please state any competing interests: Non declared

Please leave your comments for the authors below

The authors should be congratulated for their prompt review of the manuscript. They modified most of the areas of concern. The paper is now more complete and interesting.

However, I realized that my comments have not always been clear. I am sorry I was not able to correctly explain the meaning of my observations. This is particularly true in the methods section.

In general, when I expressed a concern in this section, it was not due to a doubt about the results of the study or that I had concerns about its safety, but I simply meant that that characteristic (required by the CONSORT guideline) was not clearly expressed in the methods section.

For example:

1. "Measurements for surgical safety are, also, described in generic terms ("post-operative complications, serious adverse event reporting and wound diaries") and not as an appropriate variable (for example: morbidity, total postoperative complication rate, operation specific morbidity rate, infection rate...)": The comment was not conveying the idea that measures of surgical safety were not adequate, and of course you correctly reported the outcomes required by the Monitoring Data Committee. I only pointed out that to a correct report of them did not correspond a specific notation in the methods section. The authors could increase the value of their paper by specifying the variable adopted for the outcome measurement instead of (or in addition to) the font of data they used. In particular, in the result section, the outcome is expressed in terms of "serious event rate" and "wound complication rate"; because the authors have now well explained the definition of serious event, in the methods section they could write "Early surgical safety was assessed in terms of serious event and wound complication rates" or "Early surgical safety was assessed using post-operative complication reports, serious adverse event reporting and wound diaries, as wound complication and serious event rate".

We have now revised the wording to read 'Early surgical safety was assessed in terms of serious adverse event and wound complication rates'

2. "Item 7a of the cited CONSORT guidelines recommend that the rationale for the numbers chosen for the feasibility trial be specified in the methods section; the manuscript "sample size" section specifies only the recruitment aim and not its rationale. [...] In this particular case, it could be related to the ability to recruit the sample proposed for the main trial (800, as briefly mentioned only in the discussion) in a certain amount of time." Of course, as you specified in your reply, the HART trial management group felt that the sample size for the feasibility trial was adequate. But this aspect needs to be detailed in the methods section. For example: "The feasibility trial aimed to recruit a total of 30 patients over a 5-month period, because the HART trial management group felt that such a sample size was indicative of the ability to recruit the sample proposed for the main trial within the established time frame"

The following wording has now been added to the methods section 'The feasibility trial aimed to recruit a total of 30 patients over a 5-month period, because the HART trial management group felt that such a sample size was indicative of the ability to recruit the sample proposed for the main trial within the established time frame.'

3. "the mechanism to produce the random sequence should be specified": besides the randomization design you should mention the mechanism to produce the random sequence (i.e. closed envelopes, randomization utility on a computer, coin toss). Also who generated the sequence (a surgeon, a statistician, a nurse?) and who assigned the patient to the study (it seems the main surgeon after having seen the operative picture) are useful details here. These apparently trivial details greatly enhance the value of the report.

The following wording has now been added to the methods section 'An adaptive randomisation design was used to allocate eligible patients to groups of similar size; This randomisation is based on an independent, computer-based sequence, generated from an implementation of the dynamic algorithm, using operation category (elective or emergency) and surgeon as stratifying variables [16].'

4. In the results section: "Also, dates of recruitment and follow-up (item 14a) are missing": "This has been added to the results section". The data of recruitment has been added. Still, I cannot find if there was a minimum follow up period granted to all patients and the average follow up for each group. A mention of patients eventually lost to follow up would also be nice. Of course, a trial significance is as good as its follow up rates, and adding these data would also have a positive impact on the readers.

We have added the following wording to the methods section 'Follow-up will continue for 5 years post-operatively, however in this paper, only 12-month lost to follow-up data will be presented as the aim of this feasibility trial is to assess the deliverability and safety of the trial.'

In conclusion I think that the manuscript can be accepted for publication on BMJ Open, and I also think that adding those minor details to the final text would enhance the value of the paper.